# Serial high-sensitivity cardiac troponin testing for the diagnosis of myocardial infarction: a scoping review

Hirotaka Ohtake,[1] Teruhiko Terasawa ,[1] Zhivko Zhelev ,[2] Mitsunaga Iwata,[1] Morwenna Rogers ,[3] Jaime L Peters ,[4,4] Chris Hyde[5]

¹Department of Emergency and General Internal Medicine, Fujita Health University, Toyoake, Aichi, Japan
²University of Exeter Medical School, University of Exeter, Exeter, UK
³NIHR CLAHRC South West Peninsula, University of Exeter, Exeter, UK
⁴Peninsula Technology Assessment Group (PenTAG), University of Exeter, Exeter, UK
⁵Exeter Test Group, University of Exeter, Exeter, UK

**Correspondence to**
Dr Teruhiko Terasawa;
terasawa@fujita-hu.ac.jp

## ABSTRACT

**Objectives** We aimed to assess the diversity and practices of existing studies on several assays and algorithms for serial measurements of high-sensitivity cardiac troponin (hs-cTn) for risk stratification and the diagnosis of myocardial infarction (MI) and 30-day outcomes in patients suspected of having non-ST-segment elevation MI (NSTEMI).

**Methods** We searched multiple databases including MEDLINE, EMBASE, Science Citation Index, the Cochrane Database of Systematic Reviews and the CENTRAL databases for studies published between January 2006 and November 2021. Studies that assessed the diagnostic accuracy of serial hs-cTn testing in patients suspected of having NSTEMI in the emergency department (ED) were eligible. Data were analysed using the scoping review method.

**Results** We included 86 publications, mainly from research centres in Europe, North America and Australasia. Two hs-cTn assays, manufactured by Abbott (43/86) and Roche (53/86), dominated the evaluations. The studies most commonly measured the concentrations of hs-cTn at two time points, at presentation and a few hours thereafter, to assess the two-strata or three-strata algorithm for diagnosing or ruling out MI. Although data from 83 studies (97%) were prospectively collected, 0%–90% of the eligible patients were excluded from the analysis due to missing blood samples or the lack of a final diagnosis in 53 studies (62%) that reported relevant data. Only 19 studies (22%) reported on head-to-head comparisons of alternative assays.

**Conclusion** Evidence on the accuracy of serial hs-cTn testing was largely derived from selected research institutions and relied on two specific assays. The proportions of the eligible patients excluded from the study raise concerns about directly applying the study findings to clinical practice in frontline EDs.

**PROSPERO registration number** CRD42018106379.

## STRENGTHS AND LIMITATIONS OF THIS STUDY

⇒ This is the first scoping review that explored the diversity of study methodologies adopted in studies on testing with serial high-sensitivity cardiac troponin measurements in patients with suspected non-ST elevation myocardial infarction in the emergency department.
⇒ The review method included a comprehensive literature search, duplicate assessment of eligibility, double extraction of data and descriptive synthesis through tables and graphs.
⇒ The assessed specifications included the regions and sources of studies, targeted participants and their study flow, troponin assays, sampling algorithms and direct comparisons thereof, and definitions of outcomes.
⇒ The quantitative results about the accuracy and clinical outcomes were not assessed because this was beyond the scope of the review.

## INTRODUCTION

Acute coronary syndrome (ACS) is considered a major cause of death worldwide.[1 2] Acute myocardial infarction (AMI) is a form of ACS, which represents permanent cellular damage in the affected myocardium due to ischaemia. AMI is clinically subcategorised into ST-segment elevation myocardial infarction (STEMI) and non-STEMI (NSTEMI). Each type of AMI has a unique prognosis, and their managements differ substantially. Since STEMI is an acute life-threatening condition, prompt reperfusion therapy is essential. In contrast, because the prognosis of NSTEMI varies depending on its aetiology, accurate diagnosis and risk stratification based on medical history, ECG findings and cardiac biomarker concentrations are of paramount significance in patients suspected of ACS.[3 4]

For the clinical management of NSTEMI, cardiac troponin (cTn) has been used as the mainstay of clinical diagnosis since 2000.[5–8] To avoid unnecessary hospital admissions and expedite the diagnostic process, high-sensitivity cardiac Tn (hs-cTn), a group of more sensitive cTn assays,[9] has been introduced into clinical practice since 2010. Although several primary studies and meta-analyses on the single measurement of hs-cTn reported their high sensitivity and specificity,[10–12] several challenges persist. First, blood concentrations of hs-cTn troponin take

2–3 hours to increase, and they may not be detectable within 3 hours from the onset of AMI.[3 13] Second, despite its high sensitivity, elevated concentrations of hs-cTn are observed in several clinical conditions other than AMI, including acute myocardial injury (eg, acute heart failure and tachyarrhythmia) and chronic myocardial injury (eg, structural heart disease and chronic heart failure).[3 8 13] To differentiate these conditions, serial measurements of hs-cTn, that is, assessing the absolute and/or relative changes of repeated measurements, were proposed to increase the specificity for diagnosing acute MI. Based on several studies on serial hs-cTn testing algorithms with high sensitivity and high negative predictive value, the current clinical guidelines on NSTE-ACS recommend serial measurements of hs-cTn at presentation and after 1–3 hours.[3 4]

However, the comparative effectiveness of management strategies based on serial hs-cTn measurements has not been fully elucidated, because several alternative assays are clinically available and existing reports are from studies with different designs and inconsistent testing algorithms. Thus, this study aimed to explore the diversity of the methodologies used in primary studies on serial hs-cTn measurements in patients suspected of having ACS in the emergency department (ED). We constructed an evidence map of existing studies on serial hs-cTn testing for diagnosing NSTEMI and predicting 30-day clinical outcomes. We critically appraised the currently available evidence and highlighted the issues that need to be addressed in future research.

## METHODS

This study is a focused analysis performed in conjunction with a registered systematic review project (PROSPERO registration number CRD42018106379). The protocol for the original systematic review is available at https://bmjopen.bmj.com/content/9/3/e026012.long.[10] This report followed the Preferred Reporting Items for Systematic reviews and Meta-Analyses (PRISMA) Extension for Scoping Reviews.[14]

### Data search

We searched Ovid MEDLINE, EMBASE, Science Citation Index and Cochrane Database of Systematic Reviews for studies published between 1 January 2006 and 17 November 2021 with no restrictions of language or publication status. The search terms included "chest pain", "acute coronary syndrome," "myocardial infarction," "cardiac troponin", "emergency room" and their synonyms.[10] The full search strategy is available in online supplemental appendix. We excluded editorials, letters, comments, conference abstracts, review articles and meta-analyses. Also, we excluded studies assessing clinical prediction rules (eg, Global Registry of Acute Coronary Events (GRACE) Risk Score[15]).

### Study eligibility

We included prospective and retrospective studies that evaluated patients aged ≥18 years who were suspected

of having NSTEMI in an ED and had two or more serial troponin measurements using an hs-cTn assay. Eligible were studies that reported the diagnostic accuracy of AMI and/or 30-day clinical outcomes. We only included single-gate design studies, that is, studies that consisted of a single group of subjects based on a single eligibility criteria.[16] Studies that included mixed populations of patients—with suspected STEMI and NSTEMI—were included only when data for the patients with suspected NSTEMI was separately extractable. Studies that exclusively assessed patients with suspected STEMI were excluded. Two investigators double-screened the titles and abstracts and examined the full-text articles to assess eligibility. We defined hs-cTn as assays that satisfied the requirements of the International Clinical Federation of Clinical Chemistry and Laboratory Medicine (ie, <10% coefficient of variation at the 99th percentile and ≥50% measurable concentrations above the limit of detection for both males and females).[9] Discrepancies were resolved by consensus.

### Data extraction

The following data were extracted: (1) publication and study characteristics: authors, journal name, publication year, enrolment years, number of eligible and included patients, study design, the name of the study cohort(s), geographical region(s), participant age and use of ECG to exclude patients; (2) test characteristics: assays, the timing of blood sampling, cut-off values, algorithms adopted (binary testing algorithms for ruling out MI vs three-strata testing algorithms for stratifying patients into three different risk groups, ie, high-risk, intermediate-risk and low-risk groups commonly referred to as 'rule-in,' 'observational zone,' and 'rule-out' for MI diagnosis) and (3) reference standard characteristics: specific diagnostic criteria of MI, such as those defined in clinical guidelines and/or versions of the universal definition of MI, and the assessors of the final diagnoses.

### Operationalisation

Our target population was a group of patients suspected of having NSTEMI who presented at the ED. We recorded the numbers of patients suspected of having NSTEMI who presented at the ED, patients who completed one or more study-specific serial testing algorithm(s), and patients who were assessed for test accuracy and/or 30-day clinical outcomes. A testing algorithm was specified based on the number and timing of hs-cTn measurements. The results for the algorithm were typically reported as a single value measured at presentation, together with an absolute or a relative difference between specific measurement time points (typically at presentation and a few hours later), which was categorised as delta or percent change in the hs-cTn concentrations. We classified the studies that involved two hs-cTn measurements into three groups, namely, 0 and 1 hour, 0 and 2 hours, and 0 and 3 hours, based on the blood sampling timing (in hours) of the first and second samples. Other algorithms involving

three or more blood samples were grouped into a separate category, labelled as 'others.' Assays were classified by specific manufacturers, that is, Abbott (Abbott Laboratories, Illinois, USA), Roche (Roche Diagnostics, Basel, Switzerland), Siemens (Siemens Healthcare, Erlangen, Germany) and Beckman (Beckman Coulter, California, USA). Assays by other manufacturers were categorised as 'miscellaneous.'

To assess the evidence, we used the study with the largest sample to avoid double-counting when multiple studies reported (partially) overlapping patient populations. For the study locations, we assumed that each specific research institution involved in the study assessed patients residing within its geographical region only. Comparative studies were defined as studies that adopted a paired design to assess multiple assays on the same study participants and directly compared the diagnostic accuracy for AMI or 30-day clinical outcomes. This review did not standardise the definition of AMI or the 30-day clinical outcomes and adopted the study-reported outcome definitions as specifically reported.

## Analyses
We considered each publication to be the unit of analysis and performed descriptive analyses by using percentages or medians and ranges. We combined data as a weighted average only if the pertinent data were available for specific subgroups. The assessed design specifications included the regions and sources of studies, characteristics of targeted participants and their study flow, specific troponin assays used, sampling algorithms with their operational characteristics, direct comparisons of two or more algorithms and/or assays, and definitions of the index MI and 30-day outcomes.

The volume of clinical evidence was assessed with graphs and tables using Stata V.17.0 (Stata). The graphical presentation of the study locations was constructed using Google Maps (Google, Mountain View, California, USA) and Mapcustomizer (available at https://www. mapcustomizer.com/).

## Patient and public involvement
We did not involve patients or the public in the preparation of this scoping review.

## RESULTS
### Inclusion of primary studies
Figure 1 shows the PRISMA flow diagram for this scoping review. Our search yielded 6838 articles; 6230 of them were excluded after examining the titles and abstracts. After excluding 549 perused full-text articles, we finally included 86 publications, including 72 reports on test accuracy and 52 on 30-day clinical outcomes (online supplemental appendix table 1).[17–102] The excluded articles are listed in online supplemental appendix table 2.

### Study characteristics
Eighty-three studies (97%) were based on prospectively collected blood samples, and the median sample size was

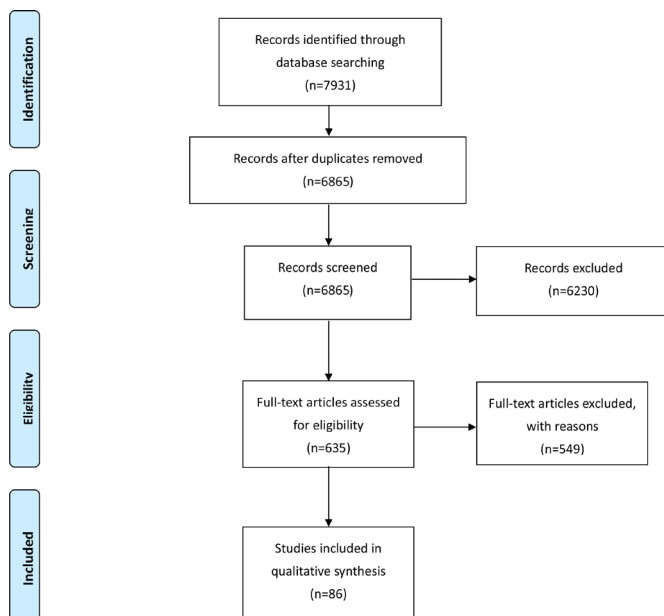

**Figure 1** The Preferred Reporting Items for Systematic reviews and Meta-Analyses (PRISMA) flow diagram.

1391 (range 160–6403) (online supplemental appendix table 1). The most commonly assessed assays were manufactured by Abbot (n=41) and Roche (n=53), followed by those manufactured by Siemens (n=10) and Beckman (n=6). Three studies assessed point-of-care assays that met the definition of hs-cTn.[76 80 102] Serial hs-cTn testing was predominantly assessed in Europe and North America and the participating institutions were limited to specific research centres (online supplemental appendix table 1, online supplemental appendix table 3 and online supplemental appendix figure 1). In contrast, only a few research centres per country from Australasia and Asia participated and provided pertinent data.

Of the 86 cohorts that reported accuracy and/or 30-day outcome data using a specific assay and a sampling algorithm, only 42 (49%) were considered unique, which included 78 606 non-overlapping patients (online supplemental appendix table 3). Of the 42 studies, 12 (29%) studies that assessed the assays manufactured by Abbott involved unique cohorts (seven, three and eight reports on the 0 and 1-hour, 0 and 2 hours, and 0 and 3 hours protocols, respectively). Similarly, data from 25 of 42 studies (60%) that assessed the assays manufactured by Roche involved unique cohorts (16, 5 and 4 reports on the 0 and 1-hour, 0 and 2 hours, and 0 and 3 hours protocols, respectively).

Nineteen studies compared two or more assays using a paired design (table 1). The most commonly reported comparison was between the assays by Abbott and Roche (17 studies on test accuracy and six on 30-day outcomes). A few studies have performed head-to-head comparisons involving the same patients. Two studies compared hs-cTn with an earlier generation non-hs cTn assay.[17 51]

Typically, the studies reported only the number of enrolled patients (ie, patients suspected of having

**Table 1** Direct comparisons of alternative high-sensitivity cardiac troponin assays*

| Hs-cTn assays | Abbott | Roche | Siemens | Beckman |
|---|---|---|---|---|
| **0 and 1-hour protocol** | | | | |
| Abbott | | 6 (10731) | 0 | 0 |
| Roche | 8 (13779) | | 0 | 0 |
| Siemens | 2 (1235) | 1 (418) | | 0 |
| Beckman | 1 (278) | 1 (278) | 1 (278) | |
| **0 and 2 hours protocol** | | | | |
| Abbott | | 2 (4063) | 0 | 1 (1118) |
| Roche | 5 (7497) | | 0 | 1 (1118) |
| Siemens | 1 (313) | 1 (313) | | 0 |
| Beckman | 1 (1118) | 1 (1118) | 1 (217) | |
| **0 and 3 hour protocol** | | | | |
| Abbott | | 1 (2945) | 0 | 0 |
| Roche | 4 (6419) | | 0 | 0 |
| Siemens | 1 (1809) | 1 (1809) | | 0 |
| Beckman | 1 (1110) | 1 (1110) | 1 (1110) | |
| **Miscellaneous** | | | | |
| Abbott | | 0 | 0 | 0 |
| Roche | 1 (1735) | | 0 | 0 |
| Siemens | 0 | 1 (830) | | 0 |
| Beckman | 0 | 1 (830) | 1 (830) | |

*For each algorithm (ie, 0 and 1-hour, 0 and 2 hours, and 0 and 3 hours protocols and a combined, miscellaneous protocol group described in the left-most column), cells in the lower left and upper right parts (separated by a right-lower diagonal line comprising black closed cells) show the volume of comparative evidence on diagnostic accuracy of acute myocardial infarction and 30-day clinical outcomes, respectively. Each cell represents a specific comparison between assays described in the leftmost column and the top row. The number of comparative studies that compared a specific pair of assays is followed by the number of assessed patients in parentheses.
Hs-cTn, high-sensitivity cardiac troponin.

NSTEMI who were eligible for and agreed to participate in the study), and the complete data on all clinically relevant patients (ie, the number of all patients suspected of having NSTEMI who presented at the ED) were missing. Fifty-two studies (60%) reported quantitative data on patients who failed to complete a study-specific serial testing algorithm(s) or patients whose diagnosis or 30-day clinical outcomes could not be established (online supplemental appendix table 1). Various proportions (median 20%; range, 0%–90%) of enrolled patients were excluded from the analysis, typically due to missing blood samples or the lack of a final diagnosis.

### Patient characteristics
The mean or median participant age ranged from 53 to 73 years, and most studies involved patients in their 40s–70s (online supplemental appendix table 1). Only one study[56] specifically focused on 0 and 1-hour protocol for a subgroup of elderly patients derived from three cohorts, specifically, patients aged 70 years or older. Of 86 studies, 17 (20%) excluded patients with chronic kidney disease (CKD) or those requiring regular haemodialysis. In contrast, only four studies specifically focused on 0 and 1-hour, 3 hours or 6–12 hours protocol for a subgroup of patients with renal dysfunction, which were derived from four cohorts.[61 62 64 65] Only a single study specifically focused on 0 and 3-hour protocol for a subgroup of female patients only derived from three cohorts. No further studies that focused on these sub populations have been found through the update search and manual search based on the reference lists.[52]

### Testing algorithms
Forty-seven studies (55%) assessed the three-strata algorithms. These studies stratified patients into 'rule-out (low-risk),' 'observational zone (intermediate-risk)' and 'rule-in (high-risk)' groups according to two sets of diagnostic criteria based on the concentrations of hs-cTn at baseline, 1–3 hours, and/or the difference between the hs-cTn concentrations of the two samples. Other studies conventionally categorised the patients into two strata (ie, 'rule-out (low-risk)' and 'rule-in (high-risk)' groups) based on a single set of criteria. The majority of data were based on assays manufactured by either Abbott or Roche, and the 0 and 1-hour algorithm was the most frequently reported (figure 2).

### Outcomes
Eighty-four studies (98%) adopted the universal definition of MI (versions 2007, 2012 and/or 2018)[6–8] to

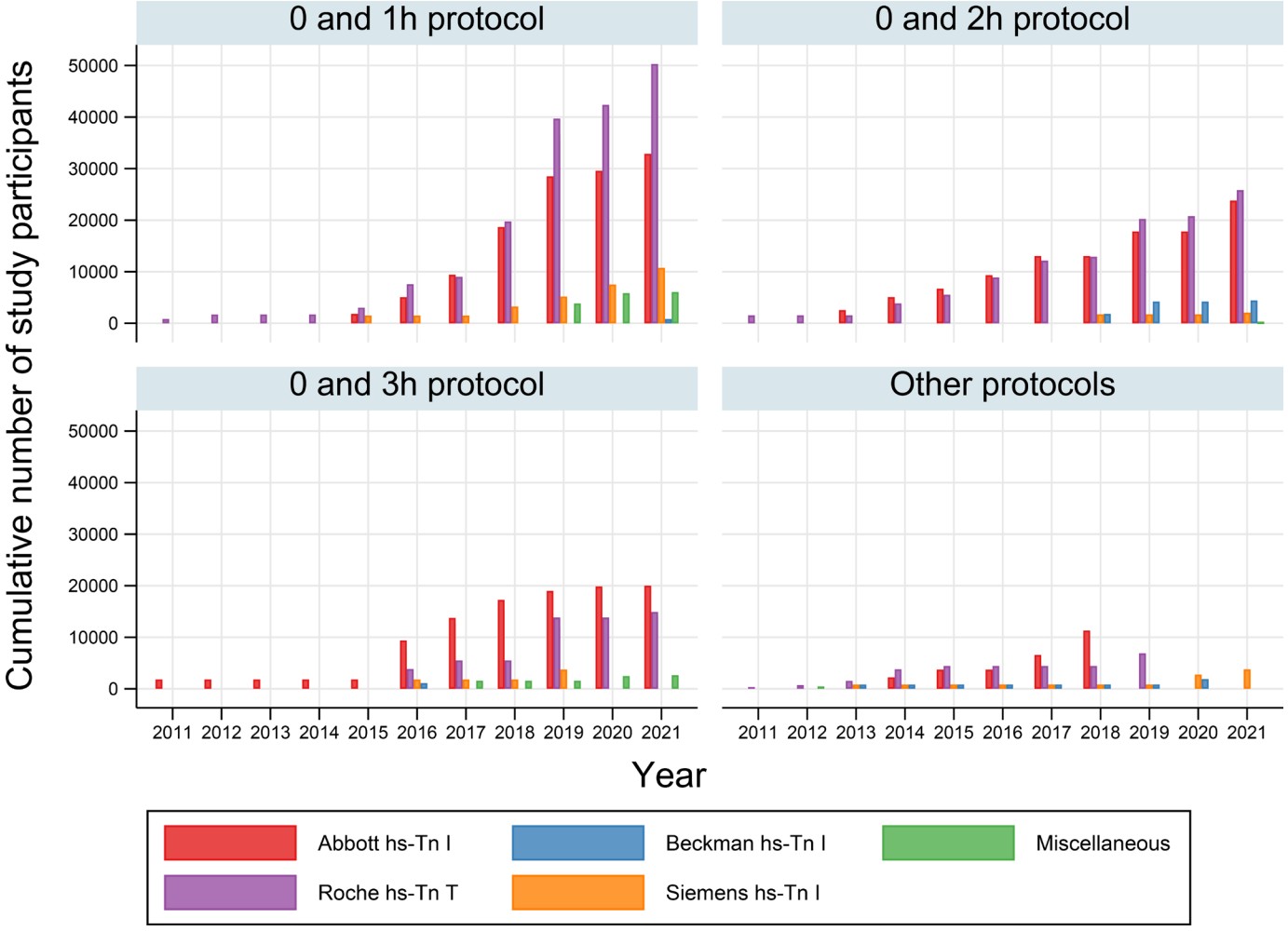

**Figure 2** Cumulative number of study participants assessed by specific testing protocols and assays for serial high-sensitivity cardiac troponin measurements*. *A number of cumulative participants are based on simple summations of all participants assessed in the relevant studies. These numbers may be overestimated due to overlapping inclusions of patients from same cohorts and/or institutions.

establish a diagnosis of AMI. A few of these studies also followed the guidelines proposed by the American College of Cardiology (ACC) (1/86, 1%),[103] the ACC and the American Heart Association (AHA) guidelines (1/86, 1%),[104] ACC and the European Society of Cardiology (ESC),[105] and the ACC/AHA[104 106] and ESC[3 107] guidelines (4/86, 5%) in addition to either version of the universal definition. Two studies[17 21] relied on the ACC guidelines[103] alone. The adjudicators of the clinical diagnosis of AMI were cardiologists in 72 studies (84%). The studies variably reported 30-day outcomes; of the 51 studies that reported one or more 30-day outcomes, 11 (22%) reported all-cause mortality and 41 (80%) reported cardiac death, whereas 36 (68%) reported major adverse cardiac events, a composite outcome including AMI, as well as cardiac death or death from all causes. Other reported clinical outcomes observed within 30 days included urgent revascularisation, percutaneous coronary intervention, coronary artery bypass graft (19/51, 37%) and ventricular arrhythmia (11/51, 22%). Twenty-seven studies (53%) also reported long-term clinical

outcomes, which included events that developed within up to 2 years.[34–37 45 57 62 65 75 94 95]

### Discussion

To the best of our knowledge, this is the first scoping review to comprehensively assess the reports on testing with serial hs-cTn measurements in patients with suspected NSTEMI in the ED. We summarised how studies measured serial hs-cTn and assessed its diagnostic accuracy for AMI and the 30-day clinical outcomes. Our results showed that most existing data were based on the Abbott cTnI or Roche cTnT assays, and the timing of the blood measurements and the diagnostic algorithms varied. The number of studies assessing these two assays using the 0 and 1-hour, 0 and 2 hours, and 0 and 3 hours protocols has been continuously increasing since 2011 when guidelines[29] recommended serial hs-cTn measurements; fewer studies have assessed other assays and/or alternative algorithms. Limited data on patients with CKD or older adults, as well as data stratified by sex, were reported,[62 64] which were deemed still under evaluation. Most studies followed the

universal definition to diagnose the index MI. However, in addition to North America or Europe, only a limited number of research teams involving several specialty institutions in specific countries have contributed to the current evidence. Most importantly, the studies excluded variable proportions of eligible patients from the analysis due to missing blood samples or concrete final diagnoses.

## Strengths

We comprehensively explored the existing evidence, focusing on how the studies were designed, analysed and reported and, the implications for clinical practice of the identified limitations and concerns. Previously reported systematic reviews focused on a two-strata rule-out strategy using unique serial measurements of hs-cTnT assay only[108] or the 0 and 1-hour three-strata strategy only[109] regardless of the assays assessed, and they did not perform a comprehensive field synopsis covering all relevant information. The objective of our scoping review is to describe the diversity in the adopted study methodologies together with their potential limitations following the standard scoping review methods. Therefore, this review should be an additional view that follows the recently published critical appraisal of the current evidence base,[110 111] both of which will help identify the current evidence gaps as well as help design future studies.

## Limitation

This scoping review performed a focused analysis on the reported study methodologies and did not address the primary objectives of the originally planned systematic review. Therefore, several limitations need to be discussed. First, we did not assess the quantitative results on accuracy and other clinical outcomes because this was beyond the scope of the present scoping review. Second, we focused on studies that assessed only test accuracy and 30-day outcomes. Therefore, data on studies that evaluated long-term outcomes have not been addressed. Third, since we excluded conference abstracts, this review may have missed newer relevant publications. Fourth, the recently developed clinical prediction rules (eg, History, ECG, Age, Risk factors and Troponin Score,[112] Emergency Department Assessment of Chest Pain Score[113] and GRACE Risk Score[15]) have consistently incorporated hs-cTn concentrations as a component variable.[40] We excluded these studies that used hs-cTn as part of risk prediction models. Finally, several studies have reported that higher hs-cTn concentrations were associated with increased mortality in patients with suspected ACS; however, our review did not address this association.[34 36 37 45 57 62 65]

## Clinical implications

Recently, a pooled study conducted by Neumann *et al* proposed a high-accurate risk-assessment tool for NSTEMI using a serial hs-cTn measurements based on an international consortium of prospectively registered data from 15 cohorts. Of these, nine cohorts were also assessed in this scoping review, at least some portions of which should have contributed to the dataset deriving the risk-assessment tool. Given the methodological concerns raised in this review, depending on how the study methodologies in the rest of the data have been addressed and/or improved, uncertainties still exist in its individualised application in real-life clinical practice. First, ACS and MI are common in elderly populations as well as patients with CKD,[114 115] while the eligible studies in our review mainly focused on middle-aged populations, and approximately one-third of the studies excluded patients with impaired renal function. In addition, hs-cTnI concentrations are generally higher in men than in women,[116] which bases the use of sex-specific cut-offs recommended by the fourth universal definition of MI in contrast to the uniform cut-off adopted in the risk-assessment tool.[117] The recent critical appraisal report[111] is also in line with this individualised approach. Moreover, quality specifications of the assays are also relevant. For example, critical factors, which are appropriate for collecting and measuring samples and applying the results to clinical practices.[118] Therefore, the optimal sample timings and cut-off values need to be validated on an individualised basis and account for age, sex and renal function under the appropriate quality control.[119 120] Second, our review found that the existing data were largely based on two assays by two manufacturers (ie, assays manufactured by Abbott and Roche); the evidence is sparse for the others. Furthermore, evidence on hs-cTnT and hs-cTnI has limited data concerning direct comparative studies of assays. Therefore, comparative studies are needed since systematic differences between hs-cTnT and hs-cTnI as well as among hs-cTnI methods have been reported.[121] Third, most of the included studies were conducted in specialised centres in Europe, North America or Australasia. A recent meta-analysis of diagnostic accuracy that focused on only the 0 and 1-hour algorithm[109] pointed out that sensitivity was not universally high across cohorts, as reported in the primary studies in these specialised centres; reproducibility of the excellent results appeared to be limited for the studies from Asia. Given this observation, validation in these regions is required. Fourth, the studies included in our review missed variable proportions of clinically relevant patients who presented at EDs with suspected NSTEMI. This appears to stem, at least in part, from convenience sampling. The failure to apply gold-standard tests to all participants may also have been responsible for the excluded cases without an established diagnosis of the cause of chest pain, which is inevitable in real-life clinical settings. These methodological weaknesses would have distorted, at least to some extent, the typical disease spectrum of clinically relevant populations. Our review failed to address how this patient loss affected the study results.

## CONCLUSIONS

Data on diagnostic test accuracy and short-term outcomes by serial hs-cTn measurements were largely derived from

particular research institutions in Europe, North America or Australasia and based mainly on two specific assays. The exclusion of variable proportions of eligible patients, which was inevitable even in well-conducted prospective studies, raised concerns regarding the studies' generalisability and direct applications in real-world ED clinical practice.

**Acknowledgements** The authors thank Drs Chihiro Kato and Jun Shinohara for assisting with data extraction.

**Contributors** TT and ZZ lead the protocol development. HO, TT, ZZ, MI, MR, JLP and CH drafted and revised the protocol. MR performed the literature searches. HO, TT, ZZ, JLP determined the eligibility of primary studies and acquired the data. HO and TT analysed the data. HO, TT, ZZ, MI, MR, JLP and CH interpreted the findings. HO and TT drafted the first version of the report. HO, TT, ZZ, MI, MR, JLP and CH critically read the manuscript and provided feedback for revision. HO, TT, ZZ, MI, MR, JLP and CH read and approved the final manuscript. HO and TT are the guarantors of this scoping review.

**Funding** This study was supported in part by a research grant from the Ministry of Education, Culture, Sports, Science and Technology (MEXT) of Japan (18K08902) and by the National Institute for Health Research (NIHR) Collaboration for Leadership in Applied Health Research and Care South West Peninsula (NIHR CLAHRC South West Peninsula; grant no, N/A).

**Disclaimer** The funders had no involvement in the study design, data collection and analysis, decision to publish, or preparation of the manuscript. The views expressed in this publication are those of the author(s) and not necessarily those of MEXT in Japan, the National Institute for Health Research, or the Department of Health and Social Care.

**Map disclaimer** The inclusion of any map (including the depiction of any boundaries therein), or of any geographic or locational reference, does not imply the expression of any opinion whatsoever on the part of BMJ concerning the legal status of any country, territory, jurisdiction or area or of its authorities. Any such expression remains solely that of the relevant source and is not endorsed by BMJ. Maps are provided without any warranty of any kind, either express or implied.

**Competing interests** None declared.

**Patient and public involvement** Patients and/or the public were not involved in the design, or conduct, or reporting, or dissemination plans of this research.

**Patient consent for publication** Not applicable.

**Ethics approval** Since it was a secondary analysis of publicly available data, no ethical review was required.

**Provenance and peer review** Not commissioned; externally peer reviewed.

**Data availability statement** All data relevant to the study are included in the article or uploaded as online supplemental information.

**ORCID iDs**
Teruhiko Terasawa http://orcid.org/0000-0002-0975-391X
Zhivko Zhelev http://orcid.org/0000-0002-0106-2401
Morwenna Rogers http://orcid.org/0000-0002-6039-238X
Jaime L Peters http://orcid.org/0000-0003-1778-3518

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
