## [Reviewer comments · BMJ Open]

ARTICLE DETAILS

TITLE (PROVISIONAL)	Serial high-sensitivity cardiac troponin testing for the diagnosis of myocardial infarction: A scoping review
AUTHORS	Ohtake, Hirotaka; Terasawa, Teruhiko; Zhelev, Zhivko; Iwata, Mitsunaga; Rogers, Morwenna; Peters, Jaime; Hyde, Chris

VERSION 1 – REVIEW

REVIEWER	Aldo Clerico Scuola Superiore Sant'Anan
REVIEW RETURNED	17-Aug-2022

GENERAL COMMENTS	To the Authors General Considerations The aim of this meta-analysis is to assess the diversity and practices of existing studies on several assays and algorithms for serial measurements of hs-cTnI and hs-cTnT for risk stratification and the diagnosis of MI and 30-day outcomes in patients suspected of having NSTEMI using a scoping review method. This meta-analysis has taken into account 86 publications, including data mainly from research centers in Europe, North America, and Australasia. Two hs-cTn assays, manufactured by Architect Abbott for hs-cTnI (43/86) and ECLIA Roche for hs-cTnT (53/86), dominated the evaluations, but also Acces Beckamn-Coulter and ADVIA Siemens for hs-cTnI were used. Only 19 studies (22%) reported on head-to-head comparisons of alternative assays. Authors concluded that: “1. Evidence on the accuracy of serial hs-cTn testing was largely derived from selected research institutions and relied on two specific assays. 2. The proportions of the eligible patients excluded from the study raise concerns about directly applying the study findings to clinical practice in frontline Eds”. The aim of this meta-analysis is original and clinically relevant, the methodology of the literature search is accurate, the analysis of data is sound, and the results reported are clinically interesting and original. I completely agree with the conclusions reported by the Authors. I would like to suggest to Authors some minor points in order to further improve the scientific message of this article concerning some information on the analytical and clinical performances of hs-cTnI and hs-cTnT methods. It is very important to point out that there are large systematic differences not only between the measured concentrations of biomarker levels between the hs-cTnI and hs-cTnT methods, but even among the 3 hs-cTnI methods (i.e. Architect Abbott, Access Beckman-Coulter and ADVIA Siemens), especially between the ADVIA Siemens in respect to the other two hs-cTnI methods. These differences can introduce some heterogeneity in the data analysis.
--

	Minor Points 1. Authors should clearly report and briefly discuss the relevance of the quality specifications for the hs-cTn assays according to the two fundamental criteria reported by 2018 by AACC and IFCC expert document (Wu AHB et al. Clin Chem 2018;64:645-655). The reference (9), reported by the Authors, is only a list of possible hs-cTn methods, without a detailed discussion on the quality specifications required for the hs-cTn methods and their clinical relevance for the diagnosis of ACS-NSTEMI. 2. Correctly, Authors point out that evidence on the accuracy of serial hs-cTn testing was largely derived from only two hs-cTn assays. Indeed, large systematic differences have been reported not only between the measured concentrations of biomarker levels between the hs-cTnI and hs-cTnT methods, but even among the 3 hs-cTnI methods (for a head-to-head comparison of analytical performances and measured concentrations in healthy subject and patients of the 3 hs-cTnI methods see: Clerico A et al. Clin Chim Acta 2019;496:25-34). Accordingly, some studies reporting the head-to-head comparisons among the results found with the most common hs-cTnI and hs-TnT are needed in order to exclude the influence of data heterogeneity among hs-cTnI and hs-cTnT methods in a meta-analysis.
--	---

VERSION 1 – AUTHOR RESPONSE

Reviewer: 1

Dr. Aldo Clerico, Scuola Superiore Sant'Anan

Comments to the Author:

[Comment]

General Considerations

The aim of this meta-analysis is to assess the diversity and practices of existing studies on several assays and algorithms for serial measurements of hs-cTnI and hs-cTnT for risk stratification and the diagnosis of MI and 30-day outcomes in patients suspected of having NSTEMI using a scoping review method. This meta-analysis has taken into account 86 publications, including data mainly from research centers in Europe, North America, and Australasia. Two hs-cTn assays, manufactured by Architect Abbott for hs-cTnI (43/86) and ECLIA Roche for hs-cTnT (53/86), dominated the evaluations, but also Acces Beckamn-Coulter and ADVIA Siemens for hs-cTnI were used. Only 19 studies (22%) reported on head-to-head comparisons of alternative assays. Authors concluded that: “1. Evidence on the accuracy of serial hs-cTn testing was largely derived from selected research institutions and relied on two specific assays. 2. The proportions of the eligible patients excluded from the study raise concerns about directly applying the study findings to clinical practice in frontline Eds”. The aim of this meta-analysis is original and clinically relevant, the methodology of the literature search is accurate, the analysis of data is sound, and the results reported are clinically interesting and original. I completely agree with the conclusions reported by the Authors.

I would like to suggest to Authors some minor points in order to further improve the scientific message of this article concerning some information on the analytical and clinical performances of hs-cTnI and hs-cTnT methods. It is very important to point out that there are large systematic differences not only between the measured concentrations of biomarker levels between the hs-cTnI and hs-cTnT methods, but even among the 3 hs-cTnI methods (i.e. Architect Abbott, Access Beckman-Coulter and ADVIA Siemens), especially between the ADVIA Siemens in respect to the other two hs-cTnI methods. These differences can introduce some heterogeneity in the data analysis.

[Response]

We want to thank the reviewer for their constructive comments and share our appreciation for their kind suggestions. We have revised the text to address all of the reviewer's comments.

[Comment]

Minor Points

1. Authors should clearly report and briefly discuss the relevance of the quality specifications for the hs-cTn assays according to the two fundamental criteria reported by 2018 by AACC and IFCC expert document (Wu AHB et al. Clin Chem 2018;64:645-655). The reference (9), reported by the Authors, is only a list of possible hs-cTn methods, without a detailed discussion on the quality specifications required for the hs-cTn methods and their clinical relevance for the diagnosis of ACS-NSTEMI.

[Response]

Thank you for your useful comments. We have revised the Discussion section accordingly. The revision (page 16, line20) reads;

'Moreover, the quality specifications of the assays are also relevant. For example, critical factors, which are appropriate for collecting and measuring samples and applying the results to clinical practices.¹¹⁸ Therefore, the optimal sample timings and cut-off values need to be validated on an individualised basis and account for age, sex, and renal function under the appropriate quality control.'

[Comment]

2. Correctly, Authors point out that evidence on the accuracy of serial hs-cTn testing was largely derived from only two hs-cTn assays. Indeed, large systematic differences have been reported not only between the measured concentrations of biomarker levels between the hs-cTnI and hs-cTnT methods, but even among the 3 hs-cTnI methods (for a head-to-head comparison of analytical performances and measured concentrations in healthy subject and patients of the 3 hs-cTnI methods see: Clerico A et al. Clin Chim Acta 2019;496:25-34). Accordingly, some studies reporting the head-to-head comparisons among the results found with the most common hs-cTnI and hs-cTnT are needed in order to exclude the influence of data heterogeneity among hs-cTnI and hs-cTnT methods in a meta-analysis.

[Response]

We agree with your comments. We have revised the manuscript to address the issues that you have highlighted. The revision (page16, line27) reads;

'Furthermore, evidence on hs-cTnT and hs-cTnI has limited data concerning direct comparative studies of the assays. Therefore, comparative studies are needed since systematic differences between hs-cTnT and hs-cTnI as well as among hs-cTnI methods have been reported.¹²¹'